# Four Cases of Perineal Groove—Experience of a Greek Maternity Hospital

**DOI:** 10.3390/medicina55080488

**Published:** 2019-08-15

**Authors:** Theodora Boutsikou, Vasiliki Mougiou, Rozeta Sokou, Maria Kollia, George Kafalidis, Zoi Iliodromiti, Christos Salakos, Nicoletta Iacovidou

**Affiliations:** 1Neonatal Department, Medical School, National and Kapodistrian University of Athens, 11528 Athens, Greece; 2Neonatal Intensive Care Unit, “Agios Panteleimon”, General Hospital of Nikaia, 18454 Piraeus, Greece; 3Pediatric Clinic, “Agios Panteleimon” General Hospital of Nikaia, 18454 Piraeus, Greece; 4Department of Pediatric Surgery, “Attikon” Hospital, National and Kapodistrian University of Athens, 12462 Athens, Greece

**Keywords:** perineal groove, neonate, anorectal malformations, perineum, gestation

## Abstract

Perineal groove is a well-defined clinical entity that belongs to a broader group of anorectal malformations. It is characterized by a non-epithelialized mucous membrane that appears as an erythematous sulcus in the perineal midline, extending from the posterior vaginal fourchette to the anterior anal orifice. The defect is gradually cicatrized, unless there are complications like infection, defecation disorders, trauma, and bleeding. The differential diagnosis includes several other conditions like trauma, infection, irritant dermatitis, lichen sclerosis, and ulcerated hemangioma. Since it is a rare malformation, it is often misdiagnosed and its presence often elicits unnecessary diagnostic workup and intervention. In this respect, neonatologists, dermatologists, or pediatric surgeons may under- or overestimate it. We report four cases of perineal groove out of 2250 live births at a Greek Maternity Hospital between September 2016 and April 2019. The “high” incidence of perineal groove cases in our Department allowed us to familiarize with this rare defect and minimize our clinical interventions.

## 1. Introduction

Perineal groove is a well-defined but rare entity that belongs to a broader group of anorectal malformations. It has rarely been described in the literature, with less than 30 cases published over the last four decades [1,2]. As the vast majority of pediatricians—and even pediatric surgeons—are unaware of the condition, its presence often elicits unnecessary diagnostic workup and intervention. This review aims at presenting our experience of four cases of perineal groove out of 2250 live births at a Greek Maternity Hospital between September 2016 and April 2019.

## 2. Cases Presentation

Signed written consent forms to publish their data and photographs were obtained from all the patients’ parents or legal guardians.

### 2.1. Case 1

A full-term female newborn with a birth weight (BW) of 3440 g was born at 39 weeks of gestation via cesarean section (CS) for breech presentation from a 37-year-old mother Gravida 3 Para 3 (G3P3); Apgar score was 6 and 10 at 1st and 5th minute of life, respectively. Baby was apneic at birth and received five inflation breaths, followed by good response with vigorous crying, good muscle tone and heart rate. The pregnancy was uneventful apart from a single echogenic intracardiac focus that was detected on the second trimester anatomy scan and was confirmed, both by fetal echocardiography (ECHO) at 22^+6^ weeks and a post-natal follow-up ECHO at 40 days of life, with no other abnormal findings. On the first physical examination, a midline defect was noticed on the newborn’s perineum, which formed an erythematous sulcus resembling a rupture that expanded about 1 cm from the posterior vaginal fourchette to the 12 o’clock position of the anterior anal opening. Both external genitalia and the anal orifice were intact and normally positioned. No active blood or passage of meconium via the vaginal opening was observed. The obstetrician placed four absorbable sutures in the malformed perineum, as he thought the defect represented tear of the area, and the newborn was transferred to the neonatal unit for prompt pediatric surgical evaluation. Frequent and meticulous hygiene of the affected area was maintained and a course of intravenous antibiotics was administered. The newborn remained stable throughout hospitalization with no evidence of infection. Parents refused to attend follow-up outpatient clinic (neonatal and pediatric consultation), and therefore we followed-up the case by frequent phone interviews; according to information provided by them, the infant had a full cicatrization of the perineal defect, with no defecation issues or local infection incidences.

### 2.2. Case 2

A full-term female newborn with a BW of 3540 g was born at 39 weeks of gestation via CS for breech presentation from a 35-year-old mother G1P1; Apgar score was 9 and 10 at 1st and 5th minute, respectively. On the ultrasound of the second trimester’s anatomy scan, mildly dilated lateral ventricles were detected, a finding that was confirmed by a cranial ultrasound on the 4th day of life. The first physical examination revealed a mildly erythematous sulcus extending from the posterior vaginal fourchette to the anterior anal orifice (Figure 1). The diagnosis of perineal groove was confirmed by a Pediatric Surgeon. Since no other anorectal or genital anomalies were present and the baby passed stools normally, the perineal defect was simply observed and remained stable throughout the neonate’s 5-day hospitalization. Mild improvement in the epithelization of the area was already evident at discharge (Figure 2).

### 2.3. Case 3

A full-term female neonate with a BW of 3500 g was born at 39^+4^ weeks of gestation by vacuum-assisted vaginal delivery, due to nuchal cord, from a 31-year-old mother G1P1; Apgar score was 8 and 9 at the 1st and 5th minute, respectively. The pregnancy was uneventful apart from repeated detection of positive IgM antibodies for cytomegalovirus (CMV) from 12 weeks onwards. The neonate had a normal cranial ultrasound on the 2nd day of life and urine PCR (polymerase chain reaction) for CMV was negative. At birth, the presence of a mucous membrane in the midline of perineum from the posterior vaginal fourchette to the anterior anal orifice was noted. Pediatric surgical consultation confirmed the diagnosis of perineal groove. As the streak appearance of the anal canal was missing at the 12 o’clock position due to the presence of mucosa, anal stimulation was performed, showing a concentric contraction of the anal sphincter which was intact. The vaginal opening was also normal with no fistulae (Figure 3 and Figure 4). The neonate was discharged on the 5th day of life with a plan for surgical correction of the malformation if hemorrhage or injury occurred.

### 2.4. Case 4

A term female neonate with a BW of 2930 g was born at 38 weeks of gestation by CS, due to fetopelvic disproportion, from a 31-year-old mother G1P1; Apgar score was 9 and 10 at the 1st and 5th minute, respectively. The pregnancy was uneventful. The first physical examination revealed an erythematous mucous membrane extending from the posterior vaginal fourchette to the anterior anal orifice (Figure 5). The defect was simply observed and the neonate was discharged with a reassessment plan in case of complications.

## 3. Discussion

Perineal groove is a rare congenital malformation of the perineum characterized by a non-epithelialized mucous membrane that appears as a wet sulcus in the perineal midline, extending from the posterior vaginal fourchette to the anterior anal orifice [1]. The lesion is usually a well-defined, erythematous, superficial mucosal band, replacing the normal perineal skin [2] with an upward direction from the 12 o’clock position of the anal opening towards the posterior fourchette [1]. According to Stephens et al., three criteria need to be met in order to define this malformation: (1) Wet groove; (2) hypertrophy of minoral tails; and (3) normal anatomy of both genitalia and urethra [3]. In their retrospective review, Esposito et al. presented six cases of perineal groove in their center in eight years, but in general, the incidence is unknown [4]. It seems to appear more frequently in females than males; there has only been one case of perineal groove combined with hypospadias and bifid scrotum described in a male newborn [5].

The diagnosis is set on clinical grounds, by the presence of the aforementioned lesion, as it is otherwise asymptomatic [1]. It is usually not associated with other abdominal or pelvic anomalies, as confirmed by abdominal, pelvic, and spine ultrasound. There is no improvement with the use of barrier or antifungal creams, and the malformation is not associated with bacterial or fungal infection at birth [2].

Misdiagnosis is a common phenomenon in these cases, since the differential diagnosis includes several other conditions like infection, irritant dermatitis, lichen sclerosis, ulcerated hemangioma, perianal pyramidal protrusion, and mainly trauma [2]. Biopsy indicates areas of non-keratinized squamous epithelium mixed with transitional anorectal epithelium and other histological findings like hyperkeratosis, fibrosis, and vascular dilatation [1,2]. Full imaging of the abdomen, pelvis, and spine is important in order to exclude anomalies that have been associated with perineal groove, like urinary tract anomalies [2].

Concerning its embryological origin, the groove’s location in the place of the perineal raphe indicates that it derives from abnormal midline development, which even explains the short distance between the anus and the genitalia, or the presence of bifid scrotum and hypospadias noticed in a male newborn [6]. According to Stephens et al., the perineal groove is attributed to failure in the midline fusion of the medial genital folds between the perineal raphe and the vestibule [3]. Another theory supports the hypothesis, that perineal groove may represent a remnant of the cloacal duct, or a defect in the development of the uro-rectal septum between the 5th and 8th week of gestation [6]. This septation of the cloaca by the uro-rectal septum is crucial for the perineal anatomy, since the tip of the septum will become the perineum. Hence, any defect in its formation after the anogenital division may cause the formation of a groove. Even histologically, the groove can be explained as a relic of the uro-rectal septum [6,7]. A third theory, justifying this defect in both sexes, is the failure of the development of external genitalia during the fusion of labioscrotal folds with ectoderm forming the midline raphe. When part of this merging does not take place, the raphe is replaced by the groove [8].

Typically, perineal groove remains stable and resolves spontaneously within the 1st year of life, with complete epithelization of the sulcus between 1 and 2 years of age [1]. Complications are rare, the most frequent being inflammation and infection of the area, due to bacteria colonizing the rectum, urinary tract infections, and defecation disorders such as constipation and fecal incontinence [2]. Trauma and bleeding are rare complications and are an indication for surgery. Surgical treatment is only indicated after the 2nd year of life, mainly for aesthetic reasons, in the case that full cicatrization has not yet occurred. According to Esposito et al., interrupted sutures bear a risk of rupture due to fecal contamination, and thus it is preferable to allow cicatrization via chemical glues or even mere observation [2].

## 4. Conclusions

In conclusion, within a 31-month period we had four cases of perineal groove in our perinatal institute, all in female neonates. In all cases, the medical history of the parturients, along with the perinatal history, were not associated with the respective malformation. Due to the rarity of the condition, neonatologists or pediatric surgeons may under- or overestimate it. This may result—if underestimated—to a higher risk of infection due to defecation disorders and injury or hemorrhage of the groove with minimal handling. On the contrary, with aggressive surgical treatment, it may lead to cicatrization complications. The “high” incidence of perineal groove cases in our Department allowed us to familiarize with this rare entity and minimize our clinical interventions. With appropriate counseling and follow-up, our neonates’ clinical course was uncomplicated.

## Figures and Tables

**Figure 1 medicina-55-00488-f001:**
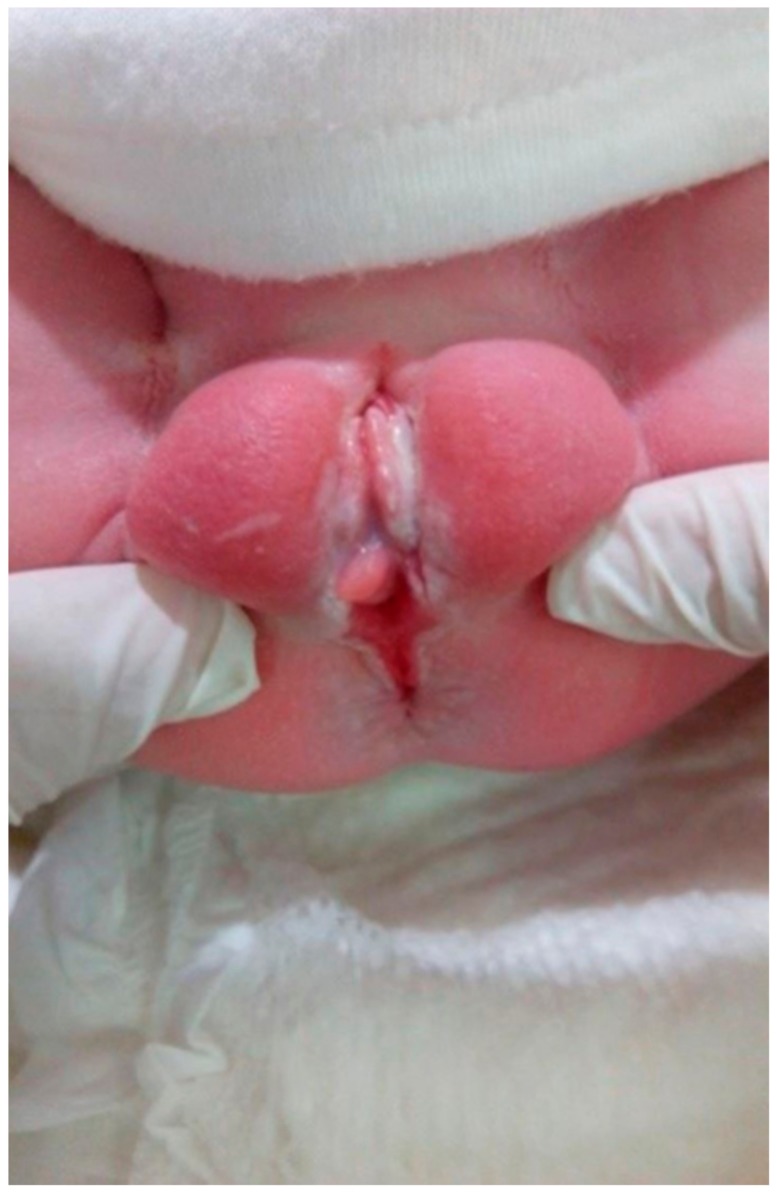
Case 2 of perineal groove.

**Figure 2 medicina-55-00488-f002:**
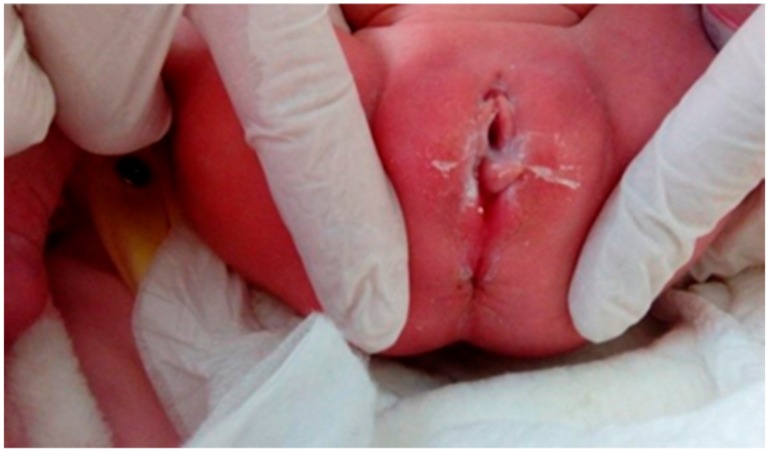
Case 2: Mild epithelization of the area at discharge.

**Figure 3 medicina-55-00488-f003:**
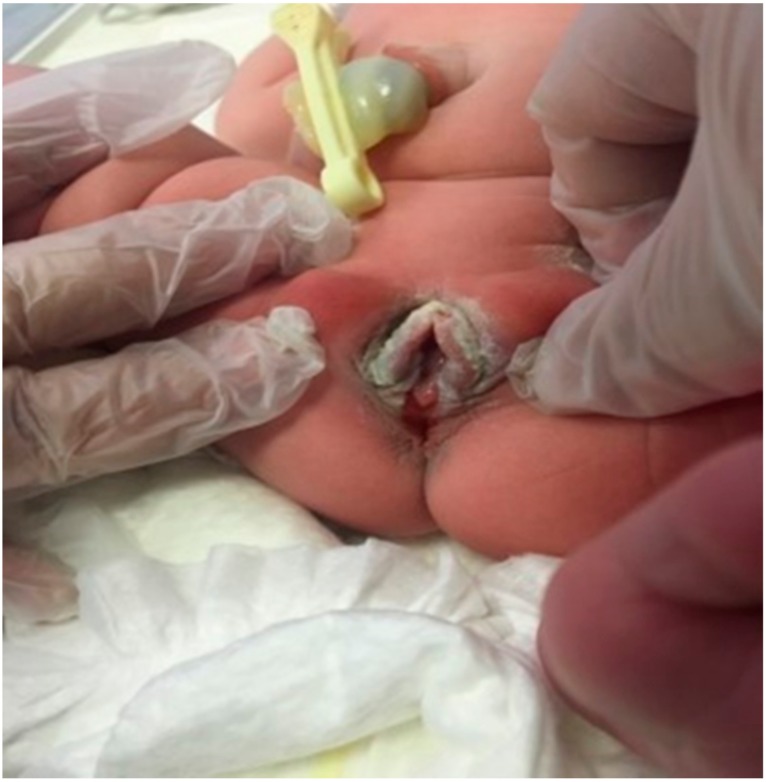
Case 3 of perineal groove.

**Figure 4 medicina-55-00488-f004:**
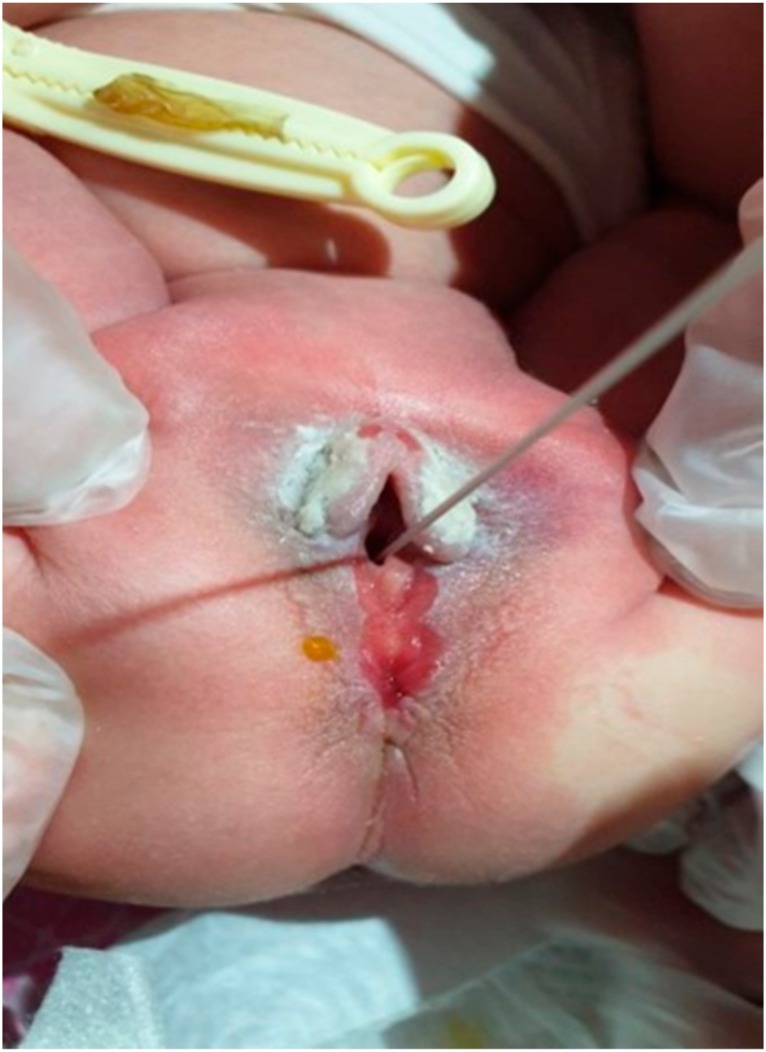
Case 3: Intact anal sphincter with normal vaginal opening.

**Figure 5 medicina-55-00488-f005:**
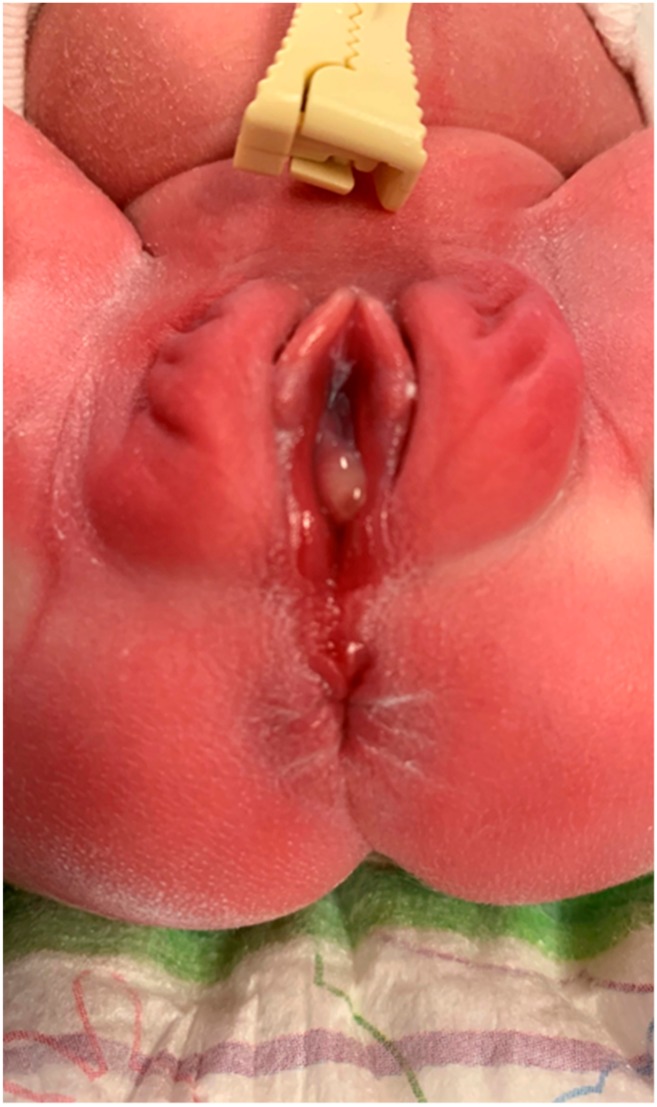
Case 4 of perineal groove.

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
