# Peer review of "Four Cases of Perineal Groove—Experience of a Greek Maternity Hospital"

_medicina, 2019, doi:10.3390/medicina55080488_

Round 1

Reviewer 1 Report

The article presented with a good review of 4 cases and extensive literatures' review. The article may need minor revision or text editing, with some suggestions as follows:

-Line 29-30: "the 30 cases sentence" may need citation/s (will be from your reference lists).

-Line 40-42: "is the single echogenic intracardiac focus", is this case has post-natal follow-up ECHO? 

-Line 80-82: "surgical removal of the malformation" may be more appropriate using "surgical correction of the malformation" term. "only in case of hemorrhage or injury" may need rewording for better reading flow, may consider "if hemorrhage or injury occurred'. 

-Line 88: "Full-term" may be more appropriate using "Term or Early Term" for a 38 weeks gestation age based on current ‘full term pregnancy” definition by ACOG since 2013. [attached article reference: ACOG Committee Opinion No 579: Definition of term pregnancy. Obstet Gynecol. 2013 Nov;122(5):1139-40. doi: 10.1097/01.AOG.0000437385.88715.4a. PubMed PMID:
24150030)].

-Line 88-93: font and alignment looks different from other pages.

-Line 109-110:it is otherwise asymptomatic” may need citation/s (will be from your reference lists).

-Line 110-111: “as confirmed by abdominal ultrasound”, may be need to add “pelvic ultrasound and spine ultrasound” for internal malformations surrounding lower abdomen, pelvic and spine area.

-Line 115-117: “hyperkeratosis, fibrosis and vascular dilatation” may need to include another citation, add citation from your reference order No.2 (Diaz, L et al).

-Line 117-119: “full imaging of the abdomen and the pelvis” may consider including spine ultrasound.

-Line 124-126: “Another theory supports the hypothesis ….. a defect in the development of the uro-rectal septum between the 5th and 8th weeks of gestation” this sentence may need citation/s (will be from your reference lists).

-Line 147-150: “This may result, in the first case to a higher risk of infection due to defection disorders and injury or hemorrhage of the groove with minimal handling, whereas in the opposite…..” the sentence sounds confusing, may need to reedit or rewording for delivering more clearance message to readers.

Author Response

Response to Reviewer 1 comments

Thank you for your comments:

Point 1: Line 29-30: "the 30 cases sentence" may need citation/s (will be from your reference lists).

Response 1: We have added the relevant reference according to the reviewer’s suggestion.

Point 2: Line 40-42: "is the single echogenic intracardiac focus", is this case has post-natal follow-up ECHO?

Response 2: In the respective case the post-natal follow-up ECHO performed at 40 days of life confirmed the presence of a single echogenic intracardiac focus, with no other abnormal findings. This was also added in the text.

Point 3: Line 80-82: "surgical removal of the malformation" may be more appropriate using "surgical correction of the malformation" term. "only in case of hemorrhage or injury" may need rewording for better reading flow, may consider "if hemorrhage or injury occurred'.

Response 3: We have complied with the reviewer’s suggestions and made the aforementioned corrections.

Point 4: Line 88: "Full-term" may be more appropriate using "Term or Early Term" for a 38 weeks gestation age based on current ‘full term pregnancy” definition by ACOG since 2013. [attached article reference: ACOG Committee Opinion No 579: Definition of term pregnancy. Obstet Gynecol. 2013 Nov;122(5):1139-40. doi: 10.1097/01.AOG.0000437385.88715.4a. PubMed PMID: 24150030)]

Response 4: We have made the respective corrections according to the reviewer’s suggestion.

Point 5: Line 88-93: font and alignment looks different from other pages.

Response 5: The text font and alignment have been corrected.

Point 6: Line 109-110: “it is otherwise asymptomatic” may need citation/s (will be from your reference lists).

Response 6: The respective reference was added.

Point 7: Line 110-111: “as confirmed by abdominal ultrasound”, may be need to add “pelvic ultrasound and spine ultrasound” for internal malformations surrounding lower abdomen, pelvic and spine area.

Response 7: Since the abdominal ultrasound included pelvic and spine ultrasound it was corrected in the text.

Point 8: Line 115-117: “hyperkeratosis, fibrosis and vascular dilatation” may need to include another citation, add citation from your reference order No.2 (Diaz, L et al).

Response 8: The respective reference was added.

Point 9: Line 117-119: “full imaging of the abdomen and the pelvis” may consider including spine ultrasound.

Response 9: The phrase “pelvis and spine” was included in the text.

Point 10: Line 124-126: “Another theory supports the hypothesis ….. a defect in the development of the uro-rectal septum between the 5th and 8th weeks of gestation” this sentence may need citation/s (will be from your reference lists).

Response 10: The respective reference was added.

Point 11: Line 147-150: “This may result, in the first case to a higher risk of infection due to defection disorders and injury or hemorrhage of the groove with minimal handling, whereas in the opposite…..” the sentence sounds confusing, may need to reedit or rewording for delivering more clearance message to readers

Response 11: We have complied with the reviewer’s suggestions and rephrased the respective sentence.

Reviewer 2 Report

The case series describes 4 female newborns suffering from perineal groove.  The article is well written and informative for neonatologists, pediatric surgeons, and readers of medicina.

I hope authors obtained written informed consent for this article by parents (please insert statement at line 33)

Please do not use capitalized letters for titles of references 1, 3, 4, 5, 7, 8.

Author Response

Response to Reviewer 2 comments

Thank you for your comments:

Point 1: I hope authors obtained written informed consent for this article by parents (please insert statement at line 33)

Response 1: We have obtained written informed consent for this article by the parents and added the respective statement according to the reviewer’s suggestions.

Point 2: Please do not use capitalized letters for titles of references 1, 3, 4, 5, 7, 8.

Response 2: We have complied with the reviewer’s suggestions and corrected the respective references.
